# Demographic Change and the Future of Austria’s Long-Term-Care Allowance: A Dynamic Microsimulation Study

**DOI:** 10.3390/healthcare13233175

**Published:** 2025-12-04

**Authors:** Ulrike Famira-Mühlberger, Thomas Horvath, Thomas Leoni, Martin Spielauer, Viktoria Szenkurök, Philipp Warum

**Affiliations:** 1Austrian Institute of Economic Research (WIFO), Arsenal, Objekt 20, 1030 Vienna, Austria; thomas.horvath@wifo.ac.at (T.H.); martin.spielauer@wifo.ac.at (M.S.); viktoria.szenkuroek@wu.ac.at (V.S.); philipp.warum@wifo.ac.at (P.W.); 2Faculty of Business, University of Applied Sciences Wiener Neustadt, Johannes Gutenberg-Straße 3, 2700 Wiener Neustadt, Austria; thomas.leoni@fhwn.ac.at; 3Health Economics and Policy Group, Institute for Social Policy, Department of Socioeconomics, Vienna University of Economics and Business (WU), Welthandelsplatz 1, 1020 Vienna, Austria

**Keywords:** population ageing, long-term care, long-term care allowance, projections, dynamic microsimulation

## Abstract

**Background/Objectives:** Europe’s demographic shift is putting increasing pressure on long-term care (LTC) systems and raising concerns about the sustainability of LTC financing. In this paper, we analyse Austria’s LTC system, particularly its universal long-term-care allowance (LTCA), and aim to project LTCA expenditure under different future scenarios. **Methods:** We use a dynamic microsimulation model to project LTCA expenditure under four scenarios up to the year 2080. Combining LTCA statistics with pooled data from the Survey of Health, Ageing and Retirement in Europe (SHARE), we estimate care needs and prevalence rates across all seven care allowance levels. This enables us to project both public spending and individual lifetime costs, disaggregated by sex and education. **Results:** Although total LTCA expenditure is projected to rise due to population ageing, scenario comparisons show that compositional shifts—such as higher educational attainment, which is linked to lower care needs and gains in healthy life expectancy accompanying mortality improvements—can significantly mitigate cost growth. The projected total expenditure increases range from 29% in a scenario where increasing life expectancy—as assumed in official population projections—is neglected to 185% in a scenario accounting for rising life expectancy but no future health gains. The findings also highlight the impact of longevity and education on the distribution of individual lifetime costs. **Conclusions:** Beyond its policy implications for LTC planning, this study demonstrates the advantages of dynamic microsimulation in capturing individual-level heterogeneity, offering a significant improvement over traditional macrosimulation approaches.

## 1. Introduction

Ageing populations and the associated increase in long-term care (LTC) needs and costs are becoming pressing issues for policymakers in Europe. Providing reliable projections is all the more essential for anticipating future care demand, guaranteeing financial sustainability through cost control while, as stressed in principle 18 of the European Pillar of Social Rights, ensuring access to affordable, high-quality LTC [1].

Broadly speaking, there are two main approaches to projecting future demand for long-term care: traditional (cell-based) macrosimulation and microsimulation models [2,3]. The former models (here referred to as macrosimulation models)—such as the models employed in the European Commission’s 2024 Ageing Report on the projection of LTC [4]—offer valuable cross-country comparability, particularly when harmonised microdata are lacking. These models typically operate under a status quo assumption, holding age- and gender-specific benefit or care-need profiles constant over time and applying them to projected population structures. Consequently, changes in LTC demand arise solely from demographic shifts. Despite some variants introducing modest adjustments to the basic assumptions, such as healthy ageing or cost-of-dying scenarios, such an approach typically remains limited in its ability to fully reflect structural trends documented in the literature, including educational expansion [5], delayed ageing, and ongoing mortality improvements [6,7]. The importance of accounting for these and other factors is underscored by evidence on the “red herring hypothesis” [8], which suggests that population ageing alone may not be the sole driver of health and LTC expenditure growth.

According to Grossman’s theoretical model [9], individual-level socioeconomic heterogeneities in health and LTC needs, as well as the resources required to meet them, persist and are therefore likely to influence future LTC costs. Education improves individuals’ efficiency in attaining and maintaining good health, allowing them to achieve better outcomes from the same medical care and health-related behaviours [9]. This effect is thought to operate through multiple mechanisms associated with higher education, including higher income, better health literacy and health behaviours, improved working conditions, and other beneficial psychosocial resources [10,11,12]. A socioeconomic gradient in health and the need for LTC is well established in the empirical literature; however, socioeconomic patterning is thought to affect expenditures and longevity in opposite directions, since less educated individuals often incur higher current healthcare expenditures while simultaneously experiencing lower life expectancy [13]. Conversely, ongoing improvements in mortality and slower biological ageing—driven by medical progress, healthier lifestyles, and related factors—may mitigate some of the health impacts of rising life expectancy. This possibility is consistent with the concept of compression of morbidity, which posits that increases in lifespan can be accompanied by a relatively shorter period of ill health [7,14,15]. Overall, although ageing increases demand—particularly for LTC—the effect has been shown to be relatively small compared to other factors, such as medical progress, rising GDP, and broader structural trends, further highlighting the limitations of status quo-based macrosimulation models, which are commonly used in research [8].

In contrast to macrosimulation models, microsimulation models simulate micro-level units—individuals, families, or households in LTC models—allowing for a more detailed consideration of distributional factors than possible with macrosimulation models. Additionally, dynamic (as opposed to static) microsimulation models simulate changes over time and in response to context changes [3]. Therefore, they offer analytical flexibility by simulating individuals’ life courses and modelling interactions between health, demographic, and socioeconomic characteristics [2,16]. These models can cover numerous factors influencing the demand and supply of care and incorporate transitions such as retirement, widowhood, or intergenerational caregiving, allowing for detailed estimates of lifetime care needs, costs, and distributional effects. The literature involving the use of microsimulation models is still limited, as data intensity and limited comparability have historically constrained such approaches. Recent work using the Survey of Health, Ageing and Retirement in Europe (SHARE) dataset and the microWELT (www.microwelt.eu) platform, however, shows that harmonised, cross-national microsimulation is feasible. Using these models enables us to integrate individual-level factors shown to be relevant in both the theoretical and empirical literature—such as education, partnership status, and caregiving roles—while aligning with aggregate demographic projections (e.g., EUROSTAT), thereby providing more realistic and policy-relevant projections of LTC demand and expenditures.

The overarching aim of this study is to leverage dynamic microsimulation modelling to capture individual-level heterogeneity and dynamic transitions in LTC demand, providing a more detailed and flexible approach than traditional macro-level projections. More concretely, focusing on demand-side factors and using Austria as a case study, we project the evolution of expenditures and lifetime costs for the long-term-care allowance (LTCA), a cash benefit that represents a cornerstone of the Austrian LTC system. We present results disaggregated by sex and education up to the year 2080 under varying scenarios. Because we use data from SHARE and apply existing Austrian care allowance regulations (which reflect differences in the severity of care needs), we expect that our results will provide more granular insights into the effects of ageing and other policy-relevant factors on the future demand for the LTCA, for which we use projected prevalence rates across various allowance levels.

To illustrate the flexibility and capabilities of a dynamic microsimulation approach, we contrast different scenarios for the demand for LTC benefits that are well grounded in the literature. Building on our baseline scenario, we both extend and restrict the assumptions made regarding care demand. Specifically, we model a scenario resembling a basic macrosimulation approach by not accounting for educational differences in care needs and thus the positive health effect associated with the educational expansion. This allows us to directly compare the differences in forecasts between a standard microsimulation and a macrosimulation-based model that is restricted to demographic factors as drivers of change. Additional scenarios explore changes in assumptions regarding ageing processes as well as the evolution of mortality rates, highlighting the potential effects of improvements in morbidity and mortality.

Recognizing that public expenditure on LTC is influenced by a complex range of factors affecting both demand and supply, our approach provides a solid foundation for further development. It can be expanded by incorporating additional scenarios that explicitly account for varying epidemiological, demographic, socio-economic, and environmental factors as well as individual habits and variations in service provision (including preventive healthcare) [2].

### Austrian Long-Term-Care Allowance

In Austria, people in need of LTC are supported by the public sector through cash benefits (long-term-care allowance (LTCA)) and benefits in kind (nursing homes, residential care facilities, home care services). The LTCA (Pflegegeld), introduced in 1993, is a needs-based but non-means-tested benefit. It is a cash payment scaled to assessed care needs and available to people of any age. Eligibility requires an assessed need for assistance for physical, mental, or psychological impairments expected to last at least six months, with a minimum assessed need of at least 65 h of care per month. Dementia triggers an automatic 40 h care premium in the assessment, and certain conditions (e.g., blindness and paraplegia) qualify for predefined benefit levels [17].

LTCA is paid directly to beneficiaries to help cover additional care-related expenses. There are no restrictions on how recipients use the funds, and there is a legal entitlement to the benefit for those meeting the criteria. Depending on the intensity of care required, the LTCA is divided into seven levels, ranging from EUR 200.80 per month at LTCA level 1 to EUR 2156.60 per month at level 7 (as of 2025, with figures annually adjusted for inflation).

In 2023, about 484,000 people—5.2% of the Austrian population—received the long-term-care allowance (LTCA), which is financed from general federal tax revenues and amounted to EUR 3.1 billion (0.65% of GDP). About half of all beneficiaries fall into care levels 1 and 2, while only 6% are in the highest levels, 6 and 7. Older persons are particularly affected: 22% of those aged 65 and over and 53% of those aged 80 and over received LTCA, with 78% of all recipients being 65 or older. Given its scale—almost matching the EUR 3.2 billion in public spending on long-term care services—the LTCA is a key pillar of Austria’s care system and a prerequisite for accessing complementary benefits such as financial support for 24 h care or assistance for caregiving relatives. In this study, we use dynamic microsimulation to estimate the future development of LTCA expenditures.

Data show that about 40% of LTCA recipients do not receive public in-kind benefits and are instead exclusively cared for by relatives or other informal caregivers. Thus, the Austrian LTC system is characterised by a rather strong reliance on family and other forms of informal care—stronger than in the Nordic countries but weaker than in Southern European countries [18].

By the end of 2022, nearly 100,000 individuals in Austria were receiving formal home care services, while approximately 68,000 were cared for in residential settings with public co-financing. These figures represent 21% and 14% of LTCA recipients, respectively [19]. Additionally, around 29,000 LTCA recipients employed self-employed live-in caregivers, predominantly from Eastern European countries, representing roughly 6% of all LTCA recipients and receiving additional means-tested public benefits of up to EUR 800 per month.

The Austrian LTCA is deeply embedded in the country’s care regime and broader welfare architecture. At the same time, it shares several structural characteristics with cash-for-care (CfC) schemes in other European countries, making its examination relevant beyond the national level. The introduction of the Austrian LTCA in 1993 reflected a wider European policy shift, as welfare states began to establish or expand monetary benefits for people with care needs while limiting the direct public provision of long-term care services [20]. The growth of CfC schemes has become a key element in the transformation of LTC systems towards what has been described as “optional familialism through the market”, i.e., a configuration that promotes informal care while fostering care markets and offering families the choice to purchase paid support [21].

Comparative analyses show that most EU Member States now employ some form of cash benefit dedicated to long-term care [22]. These schemes differ in funding sources, benefit levels, and design features. Key dimensions include the generosity and coverage of benefits, whether they are bound or unbound (that is, whether beneficiaries are restricted in how they may use the money), and whether eligibility is universal and needs-based or means-tested [22,23]. Within this typology, the Austrian LTCA shares important similarities with the German Pflegegeld and, to a lesser degree, with Spain’s Prestación para cuidados en el entorno familiar and Italy’s Indennità di accompagnamento. All these programmes are designed as universal entitlements, granted on the basis of assessed care needs rather than income or age [23]. In each case, the allowance functions as an implicit form of support for informal caregiving, providing financial assistance without linking it directly to the purchase of care services [20].

With the exception of Italy, where the LTCA is paid at a single flat rate, all these schemes entail different benefit levels, allocating higher payments to individuals with more severe care needs. Austria’s LTCA can be counted among the most comprehensive and generous cash-based programmes in Europe [22]. Germany’s LTCA is also very progressive in terms of needs, with five benefit levels. When benefits are expressed relative to the median income among older people, the German model appears slightly more generous than the Austrian one. However, Austria provides higher absolute benefit levels and covers a larger share of the population [22,23]. The Italian LTCA has an intermediate coverage level, slightly lower than that of the Austrian but higher than the German scheme, with a uniform, more modest benefit level. In contrast, the Spanish scheme reaches a smaller share of the population and offers comparatively low support across its three benefit levels, reflecting both stricter eligibility and lower funding levels.

## 2. Materials and Methods

To project government expenditure on the Austrian care allowance, in this paper, we extend the Austrian dynamic microsimulation model microDEMS via a module for care needs. microDEMS is a detailed Austrian implementation of the comparative microsimulation model microWELT (see www.microwelt.eu, (accessed on 16 September 2025) for further information) that has been used extensively to study the interplay between demography, economy, and the welfare state [5,24,25,26,27,28] and was recently extended for the comparative study of long-term care systems [16,29]. Sharing its basic architecture with microWELT, microDEMS is an interactive population dynamic microsimulation model set in continuous time. The model uses the Austrian Microcensus from 2018 as its starting population and draws on various administrative data sources to provide detailed labour force projections. For this paper, we draw on core microDEMS processes that describe how the Austrian population will evolve according to age, sex, and education (i.e., fertility, mortality, education, and migration) and complement this by modelling care allowances for the population aged 65 and older. We restrict the analysis to individuals aged 65 and older, as the vast majority of care recipients belong to this group and the risk-of-care dependence rises steeply with age, allowing a clearer focus on age-related patterns of care demand. In our model, education is a key factor in modelling family demographics such as number of children, the likelihood of being in a partnership, and mortality. While accounting for the impact of education on these processes, the model aligns with official population projections from Statistics Austria [30] in terms of aggregate outcomes. Our key assumption in this respect is that relative risks (such as relative risks regarding mortality between education groups) persist, while in the simulation, we adjust baseline risks to meet aggregate outcomes (such as the number of deaths by age and sex). Educational change is entirely the result of composition effects, whereby younger, more highly educated cohorts replace the current elderly population. We apply currently observed transition rates (arranged by gender, parents’ education, and migration background) between school types and levels to the current and future student population. This approach is conservative in that it does not factor in any positive trends in individual-level educational transitions. However, since educational transitions are influenced by parental education (students with more highly educated parents are more likely to pursue higher education), we project ongoing educational expansion at an aggregate level (though this will eventually level off) resulting from changes in parents’ educational backgrounds. Conversely, we also assume that the currently observed differences in educational transitions according to migration status (considering place of birth and second-generation migration background) will persist over time. The effect of population heterogeneity on educational outcomes is studied in detail in [30].

The long-term care module uses a cross-sectional imputation approach. Care needs and the care mix are reassigned monthly, with no attempt to achieve longitudinal consistency at the individual level. However, as education is accounted for in all modelled processes, including mortality, the approach allows for lifetime accounting by educational level, such as the average total time spent in institutions according to birth cohort, gender, and educational level. A limitation of this approach is that it does not allow us to assess the distribution of life-time care needs or institutionalization within an education group. Also, the current model does not account for mortality differences arranged according to care needs or institutionalization. Taken together, this means that, for example, assuming that the mortality rate among people in care homes is higher, there would be more deaths in care homes (and fewer among people not living in care homes), with care places being replaced by new people entering care homes. Meanwhile, the number of people living in care homes would remain unchanged in terms of age, gender, and education.

The data source for the parametrisation of the care allowance module is Austrian care allowance statistics for 2021, providing prevalence by age (in years), sex, and care allowance level. In addition, we apply Austrian care allowance regulations to a pooled sample from the Survey of Health, Ageing and Retirement in Europe (SHARE) to quantify individual care need in hours, following an approach developed in Warum et al. [29] The Survey of Health, Ageing and Retirement in Europe [31] is a multidisciplinary, longitudinal panel study of individuals aged 50+ across European countries (and Israel), collecting harmonised data on health, socio-economic status, family, labour market, and care arrangements. It covers the relevant dimensions of elderly care for our analysis (care demand and supply) in sufficient detail). Since care allowance levels in Austria are granted based on a standardised assessment scheme that assigns hours of care need according to specific limitations (see Table 1), replicating this scheme allows us to place SHARE respondents in the Austrian care allowance system according to their estimated total care needs.

We subsequently proceed with a two-step approach to estimating care allowance parameters. First, we estimate a parameter for receiving any care allowance according to age, sex, and education (see Figure A4). We use a fully interactive logit model pertaining to having more than 65 h of care needs (the threshold for receiving any care allowance) according to age, sex, and education (see Table A1). Considering average marginal predictions, we observe statistically significant differences in predicted probabilities across all three education groups, although the difference is less significant between the medium- and high-education groups (see Table A2). Through non-linear optimisation, the resulting predicted log odds arranged by education are then aligned to the care allowance statistics. Next, we estimate a parameter for the prevalence of each care allowance level according to age, sex, and education. Due to sample size limitations, we split hours of care need in the subsample of respondents with more than 65 h into only two categories at a threshold of 120 h. Care levels 1 and 2 thus fall into the lower category, and care levels 3–7 fall into the higher category. We then once again estimate a logit model with the dichotomous care categories as a dependent variable, making age interact with sex and education (see Table A1 and Figure A5). Again turning to average marginal predictions, we observe statistically significant differences in predicted probabilities only when contrasting the low education with the medium- and high-education groups but not among the latter two groups (see Table A2). Next, we apply the predicted odds from this equation to each of the seven care levels, aligning education differentials with the prevalence of each care level (Figure A6, Figure A7, Figure A8, Figure A9, Figure A10, Figure A11 and Figure A12).

To arrive at expenditure projections, we assign care allowance payouts by care allowance level according to official values in 2023. Our projections are thus reported in constant (2023) prices and do not account for inflation. However, we note that Austrian care allowance payments have been subject to inflation indexation since 2020.

### Projecting Long-Term-Care Demand

Building on the dynamic microsimulation model detailed above, in this part of the paper, we develop and analyse four distinct scenarios for LTCA demand. We first outline each scenario before reporting the corresponding simulation results. For each scenario, we present projections of total expenditure for each of the seven care categories applied within the Austrian care allowance scheme as well as cumulative expenditures up to the year 2080. Additionally, we provide estimates of lifetime LTCA costs per person aged 65 and over, disaggregated by sex and education level, along with projected prevalence rates of each care category by the year 2080.

The *baseline scenario* (S0—see Table 1) is a prediction based on status quo assumptions for care demand, with individual receipt of care allowance (and its amount across the varying allowance levels) according to age, sex, and education held constant over time. According to population projections made by Statistics Austria (these projections are cohort-component projections arranged by single year of age and sex starting from the register-based population on the projection base date; future births are generated from age-specific fertility rates, deaths are generated from age- and sex-specific mortality rates with assumed mortality improvements, and population change is generated from assumed levels and age-sex profiles of net international migration [32]), the primary drivers of changes in both individual and aggregate receipt-of-care allowance and the classification into each benefit category are demographic shifts—particularly changes in the size and age structure of the population aged 65 and over. Associated with a growing need for LTC, these shifts are mainly driven by population ageing and increased life expectancy. The only mitigating factor in this scenario is the expansion of education. We assume that the currently observed patterns persist, that is, that individuals with higher levels of education are less likely to require care and tend to have lower care needs in hours (because of having a lower probability of receiving care allowance and a higher probability of falling into a lower-level category if needed). Increasing educational attainment confers substantial health advantages through higher income, better health literacy, healthier behaviours, better working conditions, and stronger psychosocial resources [10,11]. A corresponding increase in the share of the 65+ population with higher education levels in the future is therefore expected to have a dampening effect on care demand and associated LTCA costs.

The aim of the scenario analysis is to test the sensitivity of the results to changes in various parameters and demonstrate the capabilities and flexibility of a microsimulation approach for predicting future care demand and costs, particularly in contrast to previously used macrosimulation models [33]. Starting from our baseline scenario, we thus both expand as well as further reduce the assumptions made for the demand for LTC. By not taking into account educational differences in care needs, the *no-education-expansion scenario* (S1) follows the basic idea of a macrosimulation model focused on demographic change, as it eliminates the mitigating effect of educational expansion.

Finally, both the *slower-ageing scenario* (S2) and the *constant-mortality scenario* (S3) represent alternative outlooks for population health and care demand. While S2 projects a slower increase in both the demand for care and the severity of care needs (and corresponding allowance categories) through a decelerated ageing process, S3 is based on the assumption of constant mortality (arranged by age and sex). A comparison of S3 with S0 thus highlights the effects of ongoing improvements in mortality. Conversely, S2 assumes that individuals not only live longer but also live with better health. In this scenario, we assume that, from 2025 onwards, people aged 65 and over will age more slowly. For care-related processes, we apply a person’s “care age”, which is defined so that people 65+ age by only four years every five years, meaning they only have a birthday every 1.25 years. A person aged 70 will then have a “care age” of 69, a person aged 75 will be 73, and so on, and 90 will be the new 85, thus adjusting age to increasing life expectancy. Put differently, while the baseline scenario implicitly assumes that individuals live longer but have more years with care needs (expansion of morbidity), the comparison to S2 highlights the potential mitigating effect of health improvements despite increasing life expectancy (compression of morbidity) [7,14].

## 3. Results

This section presents and analyses the main findings from the dynamic microsimulation by contrasting the scenarios introduced in the previous section. Starting with projections based on the baseline scenario (S0), we proceed with a comparative assessment in which each subsequent scenario (S1 to S3) is analysed relative to the baseline (S0). The primary objective is to highlight differences in the projections across scenarios rather than evaluate each scenario in isolation. For instance, our interest lies not only in the absolute increase in total expenditure within a given scenario but in how each scenario potentially shifts demand relative to the baseline. For each, we provide estimates of total expenditure (see Figure 1), lifetime costs (according to sex and education) (see Figure 2), and future long-term-care allowance prevalence rates (see Figure 3). Further details can be found in the Appendix A, where we provide statistics on life expectancy and the prevalence of each care category as of today (see Figure A1 and Figure A2), in addition to the absolute number of projected cases for each scenario (see Figure A3).

The baseline scenario shows that, without further changes in health trends or educational composition, significant additional fiscal burdens on the public budget can be expected. Figure 1 reports the projected evolution of total LTCA expenditure arranged by care allowance category in all four scenarios. We aggregate monthly care allowance recipients by category to obtain yearly expenditure using 2023 care allowance rates. According to our baseline scenario (assuming demographics are as they are in the official population projections and there is a composition effect due to the expansion of education–see Table 2), total LTCA spending on people aged 65 and over is projected to increase from EUR 2.3 billion in 2018 to EUR 5.7 billion in 2080 (see the upper-left panel in Figure 1). Please note that our projections refer only to the population aged 65+ and thus do not cover the costs associated with the LTCA for the whole Austrian population (EUR 3.1 billion in 2023). The underlying number of expected cases rises from 353,000 cases in 2018 to 854,000 cases in 2080. The relative share of each care category remains very similar, with the first two categories accounting for nearly half of all cases (see Figure A3). Notably, the increase in expenditure is driven by a surge across all categories of the care allowance scheme, with a particularly high concentration of costs in categories 4 and 5. Expenditures in these categories are projected to increase two and a half times, rising from EUR 0.5 billion in 2018 to EUR 1.3 billion in 2080 for care category 4 and from EUR 0.6 billion to EUR 1.5 billion for care category 5.

The development of expenditure over time also shows clearly identifiable phases. While we are currently still in a moderate growth phase (phase 0), this will change in around 2027, with stronger growth until around 2041 (phase 1—‘Post WW II’). After 2042 and onward to around 2057, LTCA expenditure will rise sharply under current conditions (phase 2—‘baby boomers’). The effects of the drop in birthrates due to the widespread use of the contraceptive pill in the late 1960s and early 1970s will cause care allowance expenditure to grow moderately again between approximately 2058 and 2071 (phase 3—‘pill kink’). The slightly higher birth rates in the 1990s will lead to a renewed increase in growth between approximately 2072 and 2080 (phase 4—‘Roaring 1990s’).

A comparison between the *baseline scenario* (S0, upper-left panel of Figure 1) and the *no-education-expansion scenario* (S1, upper-right panel of Figure 1) demonstrates the potential mitigating effects of educational expansion. It also underscores the advantages of a microsimulation model over a macrosimulation-based approach, which is typically limited in its ability to capture this need-reducing effect. Total LTCA expenditures are projected to increase by approximately 187% in scenario 1 (S1), which can be compared to 144% in scenario 0 (S0). In absolute numbers, when accounting for population ageing without considering the potential impact of educational expansion, total expenditures nearly triple, reaching almost EUR 6.5 billion in 2080 (see the upper-right panel of Figure 1). The difference between scenario S1 and the baseline scenario S0 thus corresponds to almost EUR 0.8 billion in 2080 (+13.6%).

The lower-left and lower-right panels of Figure 1 show forecasts based on two additional scenarios, ‘slower ageing’ (S2) and ‘constant mortality’ (S3). The scenario displayed in the lower-left panel of Figure 1 applies decelerated ageing, stretching individual ageing by assuming that, due to improvements in morbidity, care needs and hours grow more slowly with age. The slower ageing scenario can thus be viewed as comparatively more optimistic, predicting lower overall expenditures than the baseline scenario S0. Total expenditure is projected to increase by 45% to almost EUR 3.4 billion by 2080, which is 40% less than in scenario S0.

The projections under scenario S3—assuming there is constant mortality and there are no gains in life expectancy—remain the lowest in terms of expenditure, with an increase of only 29% over the projection period. As reported in the lower-right panel of Figure 1, total expenditure is expected to rise from EUR 2.3 billion to around EUR 3 billion by 2080 in this scenario. This highlights that approximately 80% of the EUR 3.4 billion increase in LTCA costs projected at the baseline is the consequence of a higher life expectancy. Notably, total expenditures in S3 are expected to peak around the year 2056, with the baby-boomers reaching older ages, before gradually declining thereafter.

Education-related differences in care needs have a substantial impact on lifetime LTCA costs. Figure 2 illustrates the average lifetime LTCA costs for individuals in the 65–75 age cohort at the start of the simulation, stratified by education and sex. The upper-left panel shows the results for the baseline scenario, S0. Individuals with lower education levels face significantly higher overall lifetime costs. This disparity is evident for both sexes, but it is particularly pronounced among women. In contrast, the differences in lifetime costs between older adults with medium and high educational attainment are relatively small.

Although the average lifetime costs are only marginally higher (approximately EUR 40,000) in the scenario reflecting a typical macrosimulation model (S1) than in the baseline, the distribution of costs across educational groups differs significantly from that in the baseline scenario. In fact, the distribution reverses when the positive effects of education on the need for care are not taken into account (see the upper-left and upper-right panels of Figure 2).

While we observe a ‘pro-lower-education’ gradient in lifetime costs (with higher public expenditures for less educated older adults) in the upper-left panel of Figure 2, the opposite is evident in the upper-right panel of Figure 2, displaying total lifetime costs under S1. When educational differences in prevalence rates are not taken into account, more-educated individuals incur higher lifetime costs than less educated individuals. This is driven by longer life expectancies (and potential years in poor health), which result in higher cumulative LTCA costs. As in the baseline scenario, substantial differences by sex are reported in S1. Lifetime costs for highly educated females amount to nearly EUR 60,000, while those for equally educated males total around EUR 38,000. Yet, for both scenarios, the largest gap in lifetime costs between men and women can be observed in the group with the lowest level of educational attainment.

Finally, the lower panels of Figure 2 confirm the previously observed ‘pro-lower-education’ gradient in lifetime costs, further underscoring the predictive strength of the education composition effect as captured by the dynamic microsimulation model. Total lifetime costs are significantly higher for the less educated older adults than for those with medium or high levels of education. For all education groups, costs are projected to be lower under both the slower-ageing and constant-mortality scenarios compared to the baseline scenario (under slower ageing, this is because of healthier lives, and under constant mortality, this is because of shorter lives).

Figure 3 illustrates the number of LTCA recipients by category, age, and sex as projected for the year 2080 and contrasts it with the projected total population. The upper-left panel highlights a significant increase in the prevalence of care needs among individuals aged 65 and older in the baseline scenario, as evidenced by the rising number of care allowance recipients (for comparison with 2020, please refer to Figure A2). This increase occurs gradually with age, reaching its peak around age 88. The trend is particularly pronounced among women, although a notable rise in uptake can also be observed among men. The upper-right panel shows the projected prevalence rates of LTCA categories in S1. These projections confirm the sex-related disparities reported in the baseline (see the upper-left panel of Figure 3) and the broader differences observed between scenarios S0 and S1. There are significantly higher prevalence rates when the effect of education is not taken into account, especially for women. Lastly, the lower panels of Figure 3 illustrate the lower numbers of LTCA recipients in S2 and S3 that would result from either gains in healthy life expectancy or unchanged life expectancy.

## 4. Discussion

Demographic projections for Austria, as for many other countries, highlight the trend towards an ageing population. Statistics Austria projects that the share of the population aged 65 years and older will increase from 20% in 2023 to 29% in 2080. Consequently, alongside a decline in labour supply, many demographic-related expenditures will increase. In particular, public expenditure on long-term care (LTC) in Austria is projected to double from 1.6% of GDP in 2022 to 3.1% by 2070 according to macro-projections made by the European Commission [4].

The future development of entitlement to the LTCA and the associated expenditures carry important implications for the sustainability of the welfare state and for the distribution of social benefits among the Austrian population. To understand the challenges involved and provide a sound basis for policymaking, we require not only accurate projections but also analytical insights that can identify intervention points and support the design of effective policy strategies.

Our projections are intended to show how the number of LTCA recipients in each care level will evolve under the assumption that the population ages in line with official projections and that the current distribution of care needs by age, gender, and educational group remains constant over time. Additionally, guided by both theoretical and empirical evidence, contrasting scenarios (no education expansion, slower ageing, and constant mortality) are used to examine how changes in individual determinants and the assumptions applied affect the projections.

As expected, the baseline scenario shows a marked increase in total expenditure. Spending rises across all care allowance levels, with a particularly high concentration of costs in levels 4 and 5. Under this scenario, the number of LTCA recipients among older adults aged 65 years and older will increase from 353,000 in 2018 to more than 850,000 by 2080. Accordingly, the share of the population receiving the benefit will rise from 5.2% in 2023 to 8.4% in 2080. A comparison with a scenario that assumes constant mortality reveals that most of the cost increase in the baseline scenario is attributable to a rising life expectancy. Without this increase, the number of recipients would reach just under 527,000 by 2052 and then slightly decline to 483,000 in 2080. Our other scenarios equally highlight the sensitivity of cost projections to different key assumptions, particularly those related to changes in population health and therefore care needs. Neglecting the positive health-related effects on care needs associated with educational expansion leads to projected total expenditures that are about 14% higher by the end of the projection period than in the scenario that accounts for the mitigating effect of educational expansion. Conversely, assuming that the expected increase in life expectancy will go hand in hand with a delay in the onset of care needs along the life course leads to substantially lower costs than in our baseline (which implicitly assumes that longer lives also lead to more years with care needs). According to the corresponding scenario, total expenditures will reach only slightly under EUR 3.4 billion by 2080, and the number of LTCA recipients will increase to 570,000 persons by 2080.

Our analyses demonstrate how results vary when key cost determinants develop differently from what would be expected under a continuation of current trends combined with official population forecasts. Unlike in most macrosimulation projections, the use of a dynamic microsimulation model allows us to account for health-relevant factors beyond demographics [7,8,9,10,11,12,13]. In line with Breyer and Lorenz [8] and the “red herring hypothesis”, our results show that ongoing population ageing is expected to increase LTC expenditures but also that the rate of growth may be either accelerated or mitigated by other time-varying factors, such as health improvements driven by medical progress, educational expansion, and related developments. Incorporating these structural adjustments into the model is thus needed to provide more realistic projections and help identify appropriate policy interventions.

This study contributes to the still limited—but growing—body of research on LTC expenditures using dynamic microsimulation models. While microsimulation models have been widely applied to various aspects of health and ageing—including projections of ageing-related diseases such as Alzheimer’s and dementia [34,35,36,37,38]—their application to LTC, especially for projecting expenditures rather than only the prevalence of care needs and under a broader set of realistic scenario assumptions, is still evolving. Existing single-country studies employing microsimulation models for LTC expenditures have emerged for both European countries [39,40,41] and non-European contexts such as the United States and Japan [42,43]. In contrast, macrosimulation models are more commonly used by public authorities and rely on aggregated data. They are often applied either at the national level or in cross-country frameworks such as those used by the European Commission [4]. However, these models remain limited in their ability to account for individual-level behavioural and policy dynamics.

By focusing on Austria and employing a dynamic microsimulation model, this study makes several notable contributions. First, it provides the first microsimulation-based evidence on LTC in Austria, focusing on LTCA expenditures up to the year 2080. Second, it applies four distinct scenarios reflecting key phenomena identified in the literature—such as education expansion, slower ageing, and the compression of morbidity. This enables projections of both public spending and individual lifetime costs, disaggregated by sex and education.

Consistent with evidence from Luxembourg [39], our results highlight the importance of accounting for individual-level factors beyond demographics. A comparison of our main findings with projections for countries with comparable CfC programmes reveals similar patterns, although these studies follow different methodological approaches and focus on different time periods and outcome variables (see, for instance, [44] for France, [45] for Germany, and [46] for Italy.

More generally, our projections of the Austrian LTCA can provide benchmarks for assessing how universal LTC cash benefits balance inclusiveness, fiscal sustainability, and the interaction between informal and market-based care. A recent counterfactual study using German data found that offering beneficiaries a choice between in-kind services and a cash benefit fosters advantageous self-selection, which enhances overall welfare without increasing—and instead possibly even reducing—public expenditure on long-term care [47]. The authors argue that broadening the scope of the cash option would be welfare-improving in the German context, where such a choice has already been available for several years. Changes in the eligibility and in the generosity of CfC benefits can be an effective policy tool with which to increase the market for care supply and tap new sources of informal care. In this respect, projections of the LTC allowance can be linked to the broader question of the future demand for care and the extent to which, given current institutional set-ups, we can expect gaps in addressing these care needs.

For example, comparative research on the development of long-term care needs and potential care gaps shows similar developments for Spain and Italy, although the exact scale depends largely on the different institutional structures of long-term care [16,29]. In contrast to Austria, Spain relies more on informal care provided by people other than partners (children, relatives, and neighbours). Because of these different starting points, the same percentage increase in care needs in Spain leads to a larger relative increase in unmet hours, i.e., a larger care gap. On a similar note, a comparison between Austria and Italy has shown that the increase in long-term care costs up to 2070 is significantly lower in Italy than in Austria [29]. The main reason for this difference is that Italy’s population is older, and therefore the relative increase over the period is lower than in Austria. Despite the lower absolute increase in Italy, the risk of large unmet care needs is higher because Italy currently relies more on informal care from people other than partners and because low fertility and smaller family networks will further reduce the availability of these informal carers in the future. Under status quo assumptions, a large part of the additional care hours needed in Italy would remain unmet, resulting in a larger care gap than in Austria.

## 5. Conclusions

Dynamic microsimulation is well suited to projecting LTC expenditures where individual-level dynamics (e.g., health transitions and household changes) and policy interactions are critical. Our study benefits from the integration of various data sources and the Austrian care needs assessment system, which we used to model hourly care requirements. Nonetheless, it also has limitations. Our use of SHARE data constrains the validation of our results, particularly in terms of measuring care needs across population subgroups due to low numbers of respondents in various subgroups in SHARE. Future research could also adapt our approach to model different scenarios that provide a deeper understanding of the individual determinants of care needs and the potential effects of various policy measures.

Our modelling approach can also serve as the basis for future work aiming to overcome some of the shortcomings of traditional macrosimulation-based projections. Importantly, it helps avoid the oversimplification that results from ignoring subgroup differences and life events. In addition, it can relax static assumptions. For instance, while the EU Ageing Report includes sensitivity tests (e.g., higher life expectancy), it has less flexibility in incorporating policy changes, limiting its ability to assess the impact of LTC reforms.

Our results highlight several critical policy implications that are also valid in other European countries with similar socio-economic characteristics: Educational expansion is a key lever for reducing future care demand. The health advantages possessed by individuals with higher education, via higher income, better jobs, and healthier behaviour, accumulate over the course of their lives, resulting in better health outcomes and a lower probability of requiring intensive care during old age [10,11,12]. This paper reveals a significant difference in life-time LTC expenditures between individuals aged 65 and older with low and high levels of education. Lifetime costs are substantially higher for those with low levels of education when education and improvements in health are accounted for. For policy, this highlights the importance of not only promoting educational attainment—especially among disadvantaged groups from an early stage—but also addressing the broader social determinants of health, such as occupational health and access to preventive services, that mediate the link between education and care needs [9]. By investing in these upstream factors, policymakers can help reduce future long-term-care demand and improve equity in healthy ageing. We have shown that improvements in health can significantly reduce LTC costs, implying that healthier ageing initiatives (e.g., preventive care and chronic disease management) and effective investments in public health could delay care dependency and yield significant long-term savings.

Alongside rising health and pension costs, the described development will put enormous strain on public finances, prompting discussion on sustainable financing models to balance the generational burden. Since the development of demand can be projected over time, it makes sense to engage in longer-term financial planning that makes intergenerational justice, redistribution effects, and the combination of public funds, insurance financing, and private contributions transparent. For instance, a LTC contribution for pensions above a certain threshold could be proposed for discussion, as the literature on long-term distributional issues shows that older people have been less affected by recent crises [48,49]. Germany, which faces demographic development similar to that of Austria, finances long-term care not through general taxation but via a dedicated social insurance contribution. It has, for instance, introduced higher contribution rates for childless employees. These examples illustrate that the design of long-term care financing can be an effective lever for ensuring both fiscal sustainability and fairness across generations.

## 6. SHARE Acknowledgements

This paper uses data from SHARE Waves 1 [50], 2 [51], 3 [52], 4 [53], 5 [54], 6 [55], 7 [56], 8 [57], and 9 [58]. See Börsch-Supan et al. [31] for methodological details.

SHARE data collection was funded by the European Commission, DG RTD through FP5 (QLK6-CT-2001-00360), FP6 (SHARE-I3: RII-CT-2006-062193, COMPARE: CIT5-CT-2005-028857, SHARELIFE: CIT4-CT-2006-028812), FP7 (SHARE-PREP: GA N°211909, SHARE-LEAP: GA N°227822, SHARE M4: GA N°261982, DASISH: GA N°283646) and Horizon 2020 (SHARE-DEV3: GA N°676536, SHARE-COHESION: GA N°870628, SERISS: GA N°654221, SSHOC: GA N°823782, SHARE-COVID19: GA N°101015924) and by DG Employment, Social Affairs & Inclusion through VS 2015/0195, VS 2016/0135, VS 2018/0285, VS 2019/0332, VS 2020/0313, SHARE-EUCOV: GA N°101052589 and EUCOVII: GA N°101102412. Additional funding from the German Federal Ministry of Education and Research (01UW1301, 01UW1801, 01UW2202), the Max Planck Society for the Advancement of Science, the U.S. National Institute on Aging (U01_AG09740-13S2, P01_AG005842, P01_AG08291, P30_AG12815, R21_AG025169, Y1-AG-4553-01, IAG_BSR06-11, OGHA_04-064, BSR12-04, R01_AG052527-02, R01_AG056329-02, R01_AG063944, HHSN271201300071C, RAG052527A) and from various national funding sources is gratefully acknowledged (see www.share-eric.eu, accessed on 16 September 2025).

This paper uses data from the generated easySHARE data set [59]; see Gruber et al. [60] for methodological details. The easySHARE release 9.0.0 is based on SHARE Waves 1, 2, 3, 4, 5, 6, 7, 8, and 9.

## Figures and Tables

**Figure 1 healthcare-13-03175-f001:**
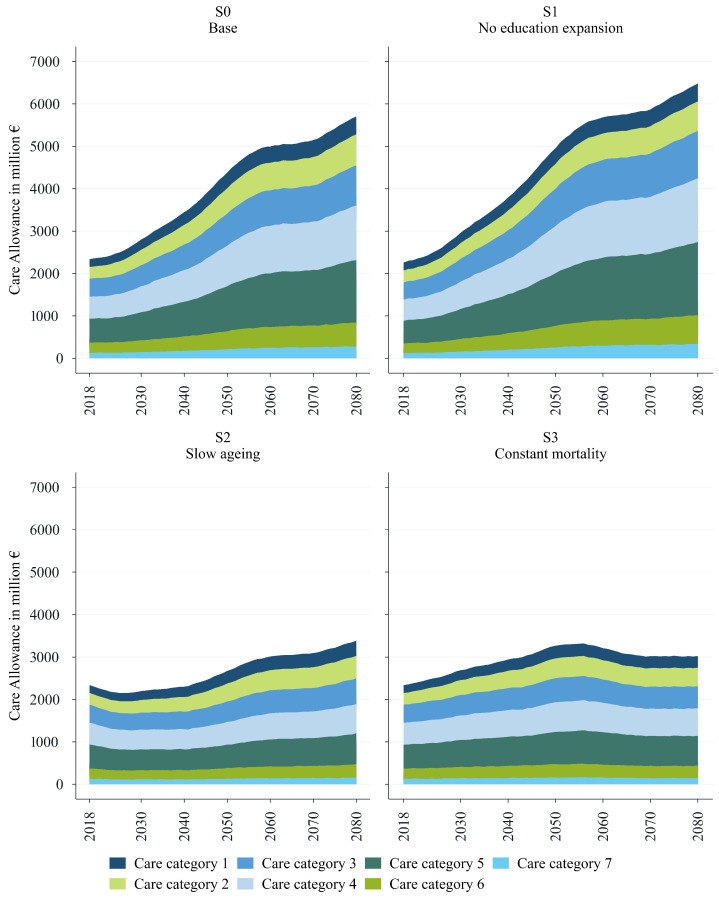
Total LTCA expenditure by category. Source: WIFO. Constant prices (2023).

**Figure 2 healthcare-13-03175-f002:**
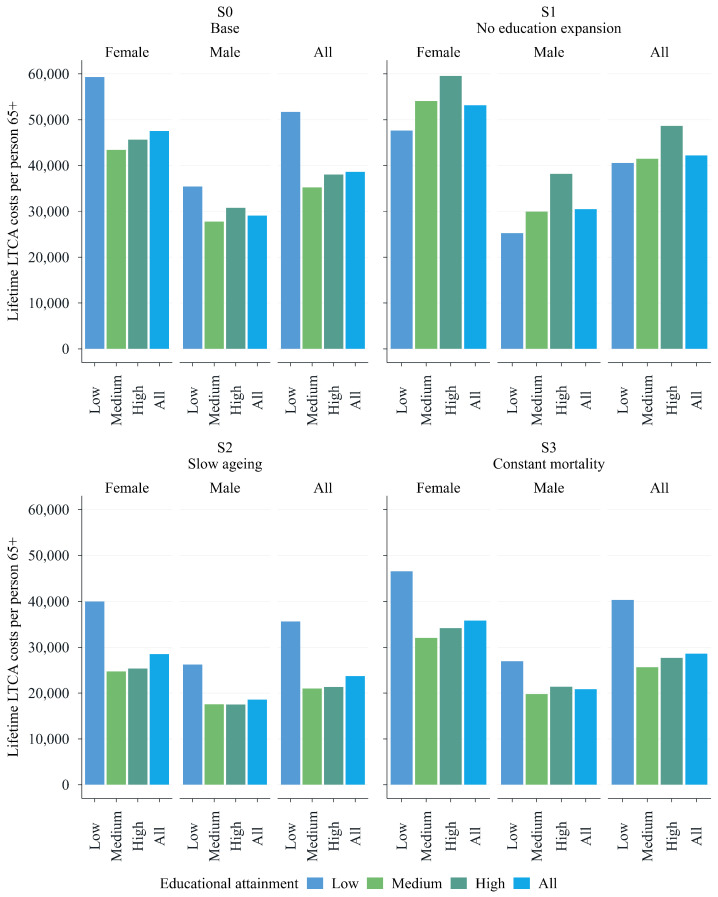
Lifetime LTCA costs per person aged 65+ (now 65) by gender and education. Source: WIFO. Constant prices (2023).

**Figure 3 healthcare-13-03175-f003:**
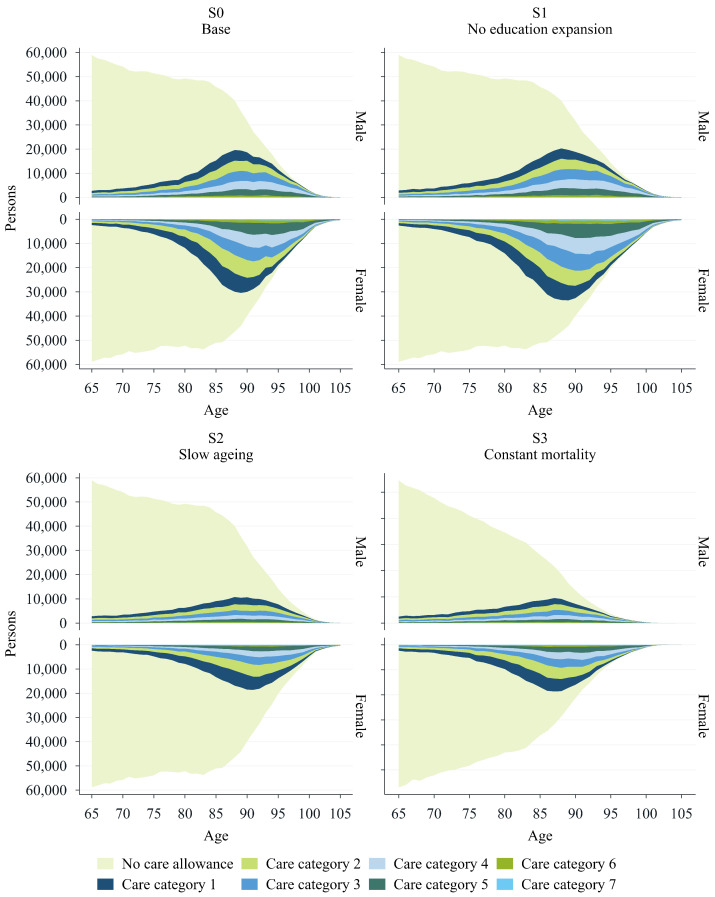
LTCA prevalence 2080. Source: WIFO.

**Table 1 healthcare-13-03175-t001:** Long-term care allowance in Austria (2025).

LTCA Level	Criteria	Amount of LTC Allowance p.m. in EUR
1	>65 h of care required p.m.	200.80
2	>95 h of care required p.m.	370.30
3	>120 h of care required p.m	577.00
4	>160 h of care required p.m.	865.10
5	>180 h of care p.m. if an extraordinary amount of care is required	1175.20
6	>180 h of care p.m. if uncoordinated care measures are required, and these must be provided regularly during the day and night, or the permanent presence of a carer is required during the day and night because there is a likelihood of danger to oneself or others	1641.10
7	>180 h of care p.m. if no purposeful movements of the four extremities with functional realization are possible or a condition requiring equal attention is present	2156.60

Source: Austrian Ministry of Social Affairs. https://www.oesterreich.gv.at/themen/pflege/4/Seite.360516.html (accessed on 16 September 2025).

**Table 2 healthcare-13-03175-t002:** Care demand scenarios.

Scenario	Name	Care Demand Assumptions
S0	*Baseline*	Demography as given in official population projections by Statistics Austria Education expansion Constant prevalence for care allowance take-up and levels by age, sex, and education
S1	*No education expansion*	Switching off differences in LTCA prevalences by education and thus the mitigating effects of educational improvements (prevalences for care allowance takeup and levels by age and sex only)
S2	*Slower ageing*	Care needs grow at a slower rate with age (a person ages 4 years in 5 years)
S3	*Constant mortality*	Mortality by age and sex as of today (no gains in life expectancy)

## Data Availability

All code files are available at https://www.microwelt.eu/ (accessed on 16 September 2025), and all data is freely available for scientific purposes from the respective sources (e.g., Eurostat and SHARE-ERIC). Data on the incidence of LTCA levels by age and gender were provided by the Austrian Federal Ministry of Social Affairs.

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
