# Peer review of "Demographic Change and the Future of Austria’s Long-Term-Care Allowance: A Dynamic Microsimulation Study"

_healthcare, 2025, doi:10.3390/healthcare13233175_

Round 1
Reviewer 1 Report
Comments and Suggestions for Authors
Report on “Demographic Change and the Future of Austria’s Long-Term
Care Allowance: A Dynamic Microsimulation Study”
The manuscript addresses an important aspect that is often overlooked in projections of future long-term care (LTC) needs. It is generally accepted that a massive increase in the number of people requiring LTC in countries such as Austria and Germany is inevitable due to the unfavorable demographic structure of both countries, characterized by very pronounced baby boomer cohorts. However, it is often neglected that population projections are based on assumptions about the future development of fertility, migration and - what is more important for LTC - life expectancy, which has a direct impact particularly on the future size of the older cohorts. In addition, forecasts of people in need of care or public expenditures for LTC very often use the status quo hypothesis, assuming constant care rates over time.
The paper compares these assumptions with other scenarios. One scenario assumes constant mortality, while another models a reduction in care needs. One innovative feature is certainly the modeling of the consequences of an increase in educational attainment on LTC needs.
The paper is well written and easy to understand, but there are some weaknesses in the numbering of the chapters. In addition, some content is duplicated.
Overall, I can highly recommend the paper for publication, but some revisions are necessary first.
Major Comments:
1. The paper tends to overemphasizes the advantages of a dynamic microsimulation model over traditional macrosimulation models. However, the use of these terms in the context of projecting diseases, health expenditures, or people in need of care is probably only familiar to a limited circle of scientists. For example, the term “macro-simulation model” is not used anywhere in the European Commission's 2024 aging report cited in the introduction. It should be clarified—as far as I understand—that a macro-simulation model primarily refers to projections that use the status quo hypothesis, i.e. constant age- and gender-related benefit profiles for health or long-term care. Some of these “traditional” methods may also model a slight shift in these profiles due to declining mortality (morbidity compression hypothesis) or a cost of dying scenario. In this regard, the limitations of traditional methods should have been explained more clearly. It is recommended to use the terms used in health economics literature here. Overall, it is advisable to embed the work in existing health economic research related to demographic transition and, in particular, the consequences of increased life expectancy on health and LTC expenditures. To this end, at least some important sources should be cited, such as those that examine the pros and cons of the “red herring hypothesis.”, i.e. Breyer, F., Lorenz, N. The “red herring” after 20 years: ageing and health care expenditures. Eur J Health Econ 22, 661–667 (2021). https://doi.org/10.1007/s10198-020-01203-x.
2. What I do not quite understand from the paper is the dynamic aspect of the modeling. Here, it should be explained more precisely how the whole thing was modeled. In particular, I would like to have the following questions answered: How exactly is education expansion modeled? Does the group with a high level of education benefit disproportionately from the increase in life expectancy? (Or) does education expansion mean that there is a higher level of education in younger cohorts compared to older cohorts, and therefore over time the level of education of the older age groups rises with the aging of the younger cohort? In other words – is there a demographic structure effect of education level? What distribution of educational attainment is assumed for younger cohorts (cohorts that are 5 years old in 2021 will be 65 years old in 2080)? Is educational attainment itself dynamic, i.e., is it modeled whether educational attainment can change for agents over time?
3. The following comment assumes that I am interpreting the microsimulation model correctly: If I understand correctly, the dynamics of the model do not include an interaction between the variables mortality and need for care, nor do they depict dynamic care careers over time. If this is indeed the case, then the model has a significant limitation which should be addressed at least in the discussion. It must be noted that, for example, if fewer people become in need of care, a year later the at-risk population will include a larger number of people. In other words, the prevalence of care is also dynamic and results over time from the development of incidence rates of new cases, transition probabilities between care levels, and the general probability of dying. It is possible (see 2.) that the probability of entering a different level of education may also be a factor. However, as far as I understand, these interactions are not captured in the current microsimulation model, but may have a significant impact on the results. This is shown by projections using Markov illness-death models, see e.g. Milan, V., Fetzer, S. & Hagist, C. Healing, surviving, or dying? – projecting the German future disease burden using a Markov illness-death model. BMC Public Health 21, 123 (2021). https://doi.org/10.1186/s12889-020-09941-6
4. A table showing the results of the logit models underlying Figures A4 to A12, particularly indicating significance levels, would enhance the paper.
5. The key assumptions underlying the population projection by Statistics Austria should be briefly mentioned.
6. Section 1.1 could be streamlined considerably. In its current form, it appears to be too long and somewhat confusing for non-Austrians. I would recommend first listing the different types of public spending on care, then providing the relevant figures (number of people, expenditure in billions and as a percentage of GDP) (or compiling them in a table), and only then discussing the institutional structure of the LTC allowance. The comments on care providers may be interesting, but they stray too far from the main topic of the paper.
7. The numbering 1.1 (which is not followed by 1.2) and 2.1 (which is not followed by 2.2) is confusing and should be revised. I cannot find chapters 4.1 and 4.2, which are referred to in 2.1.
8. Although the modeling of the shift in prevalence rates in the S2 scenario is explained in great detail (“In this scenario, we define a person’s care age such that from 65 onward, a person ages 4 years in 5 years for care-related processes (having a birthday only every 1.25 years). A person aged 70 will then have a “care age" of 69, a person aged 75 will be 73... and 90 will be the new 85, thus adjusting age to increasing life expectancy.”), it remains unclear in which year this shift took place.
9. The second paragraph of the discussion (“In this study...”) belongs in the chapter on the institutional design of the LTC allowance. The fifth section (“As expected...”) belongs in the results chapter.
10. The limitations should be explained in more detail. In particular, it remains unclear what is meant by “Nonetheless, it also has limitations. The use of SHARE data constrains the validation of our results, particularly in terms of measuring care needs across population subgroups.” What exactly is the constraint and why does it matter?
11. It is unclear what is meant by this sentence “Third, macro models have limited behavioural feedback, i.e. they lack mechanisms to model how individuals adjust their behaviour, potentially underestimating shifts in demand.” Does the microsimulation modeling used in the manuscript account for adjustments in individual behavior?
Author Response
Please find our reply to reviewer 1 attached.

Reviewer 2 Report
Comments and Suggestions for Authors
The topic of this work is extremely important and timely, and this importance will likely continue to grow. Population aging is a significant challenge for public health in various countries. The work submitted for review is extremely important and valuable. However, it lacks a proper context, both within the theory of science and within the socio-economic reality. Because it concerns only one selected country, it is worth highlighting similar reports from other European and even non-European countries. Such elements are particularly lacking in the discussion.
The introduction is quite good, but it would be worthwhile to also refer to theoretical issues and the results of other researchers. This would give the article a more scientific tone. While I am aware of the research nature of the work, the bibliography is relatively sparse in scholarly texts. The introduction and literature review do not need to be lengthy, but they should be based significantly on existing theories and results. The literature review should also provide the assumptions for the research.
Materials and methods is a section that would require a description of secondary material. This is an important element of this study, but the reader has little knowledge of how institutions collected this data. I assume such information may be available. A note regarding the description of data collection methodology used by other entities – this methodology is often provided in the studies used. However, if this is not possible, it is advisable to include this information in the text.
The discussion is practically devoid of other reports and results. It is more of a summary of the results obtained.
Conclusions should be presented in a separate section of the text.
The bibliography is not particularly extensive – the text should be more focused on theory as the basis for the discussion.
Author Response
Please find our reply to reviewer 2 attached.

Reviewer 3 Report
Comments and Suggestions for Authors
This is a well-written piece of research analyzing Austria's LTCA by the dymaic microsimulation model to project future expenditures on the social security scheme under the impact of demographic changes. The use of the dynamic microsimulation model is commendable, which sheds light on people's understanding of the long-term care needs of older adults and the upcoming demographic changes. It has also explored different forms of public expenditures on long-term care allowance based on different scenarios, which have broadened the persuasiveness of the arguments.
Yet, there are a few things that the authors can do to enhance the quality of the manuscript:
- Presentation of the findings: The results presented in the paper are sound and scientifically analyzed, which is good. However, the graphs and tables could be streamlined for easier reading. For example, there should be exclusive paragraphs for interpreting Figures 1-3, so that readers can grasp the implications of the results presented in the figures more easily. Also, can the figures in the appendix be embedded in the main text and further streamlined?
- Scenario comparisons: The analyses of different scenarios are insightful, but the visualization of the differernces in the projected expenditures due to demographic changes could be better presented. For example, using simple bar graphs?
- Although this study focuses on Austria, in order to help readers from other countries understand more about the Austrian case and the validity of your research methods, it is better to provide a cross-country comparison between Austria and other European/OECD countries, so that your analyses could be more justified.
- Policy recommendations: The study has provided insights into policy implications for Austria. However, it would be better for the authors to provide more detailed discussions on the applicability of your recommendations to other countries with similar socio-economic characteristics. Also, apart from financial aspects, what other interventions can the Austrian and other governments undertake, such as better healthcare initiatives?
Author Response
Please find our reply to reviewer 3 attached.
